# The association between BCG treatment in patients with bladder cancer and subsequent risk of developing Alzheimer and other dementia.—A Swedish nationwide cohort study from 1997 to 2019

**Eugen Wang**[1,2]*, **Oskar Hagberg**[3], **Per-Uno Malmström**[2]

**1** Center for Clinical Research, Sörmland, Uppsala University, Uppsala, Sweden, **2** Department of Surgical Sciences, Urology, Uppsala University, Uppsala, Sweden, **3** Department of Translational Medicine, Lund University, Malmö, Sweden

* eugenwang@yahoo.com

## Abstract

### Background

Alzheimer's disease (AD) affects 50 million people worldwide. The immune system plays a major role in the pathogenesis of AD. Several retrospective analyses have reported a decreased risk of AD and other dementias in bladder cancer patients treated with immunotherapy in the form of Bacillus Calmette-Guerin (BCG) bladder instillations. We tested this hypothesis in a Swedish population-based prospective cohort of patients with non-muscle invasive bladder cancer (NMIBC).

### Methods and findings

We utilized the BladderBaSe 2.0 database, which contains tumor-specific, health-related, and socio-demographic information for patients diagnosed with NMIBC between 1997 and 2019. The database also includes a matched comparison cohort sampled from the general population, consisting of individuals free from urinary tract cancer at the time of the index case's diagnosis. Five controls were randomly selected for each index case without replacement on the date of the index case's diagnosis. Our inclusion criteria identified participants diagnosed with NMIBC who had received BCG as primary treatment, along with their corresponding comparison cohort. We excluded those diagnosed with dementia before or within 6 months of NMIBC diagnosis. To compare the NMIBC cohort with their matched comparison cohort, we used a stratified Cox model, treating each case with its controls as a stratum. We identified 38,934 patients in the NMIBC cohort, with 6,496 receiving BCG after primary diagnosis (cases). AD/dementia was diagnosed during follow-up in 6.1% of cases and 7.4% of controls. Cases had a slightly lower risk of dementia than controls, with a hazard ratio (HR) of 0.88 (95% confidence interval [CI] 0.780–0.991), and a HR of 0.89 (CI 0.703–1.119) for AD. Subgroup analysis for dementia showed that age over 75 years had an HR of 0.73

**Data Availability Statement:** Data are available on reasonable request. Reports from the SNRUBC are available online (only in Swedish). Researchers can

apply for data by submitting a proposal to the BladderBaSe 2.0 steering committee and data files for studies can be uploaded to remote servers for secure analysis. For more information contact Christel Häggström (christel.haggstrom@umu.se), Umeå universitet, Medicinska fakulteten, Institutionen för folkhälsa och klinisk medicin. Department of Surgical Sciences, Uppsala University, Uppsala, Sweden.

**Funding:** This study was funded by Center for Clinical Research, Sörmland, Uppsala University and Schmekel funder for Urological research. The funders had no role in study design, data collection and analysis, decision to publish, or preparation of the manuscript.

**Competing interests:** The authors have declared that no competing interests exist.

**Abbreviations:** AD, Alzheimer´s disease; BCG, Bacillus Calmette-Guerin; BladderBaSe, Bladder Cancer Data Base Sweden; CI, confidence interval; G1/LMP, Bladder cancer Grade 1 or Low Malignant Potential; GP, general practitioner; HR, hazard ratio; ICD, International classification of diseases; NMIBC, non-muscle invasive bladder cancer; Q1-Q3, Inter quartile range, the range between the first quartile (Q1) and the third (Q3); SNRUBC, Swedish national register of urinary bladder cancer.

(CI 0.616–0.863), and female gender had an HR of 0.73 (CI 0.552–0.971). The associations were similar for AD specifically, but not statistically significant. Similar to previous studies, we analyzed bladder cancer patients treated with and without BCG therapy. Multivariate Cox analysis indicated that those treated with BCG had a lower risk of dementia (HR 0.81, 95% CI 0.71–0.92), and an HR of 0.98 (95% CI 0.75–1.27) for AD specifically. High age was a significant risk modifier; the HR was 3.8 (CI 3.44–4.11) for dementia and 3.1 (CI 2.59–3.73) for AD. Even patients not receiving BCG had a significantly lower risk for AD than controls (HR 0.86, CI 0.77–0.96).

## Conclusions

This study observed a marginally decreased risk of developing AD/dementia associated with earlier intravesical BCG treatment in NMIBC patients. This small benefit was most pronounced in those with high age and female gender. The disparity from previous highly positive studies underscores the importance of using an appropriate control cohort.

## Introduction

It is estimated that Alzheimer's disease (AD) affects 50 million people worldwide and, with increasing longevity, the prevalence is expected to be 131 million by the year 2050. Until now, there is no cure for this disease. In AD, the nerve cells in the brain atrophy, especially in the regions where memory is located, which is caused by scarring in the brain related to inflammation. The disease is characterized by the accumulation of amyloid β plaques, neurofibrillary tangles, and sustained innate neuroinflammation [1]. The reasons for the breakdown of nerve cells are not fully understood, but a process of inflammation suggests that beta-amyloid and the protein tau are involved in some way.

Immunotherapy in the form of Bacillus Calmette-Guérin (BCG) bladder instillations has been recommended for the last thirty years to reduce the recurrence of non-muscle-invasive bladder cancer (NMIBC). The exact mechanism of BCG's anticancer activity has not been deciphered, but it is well recognized that BCG manifests immune effects. It binds to fibronectin in the bladder wall and stimulates Th1 cells to secrete multiple cytokines, which induce cell-mediated cytotoxic mechanisms that eliminate cancer cells [2]. With its potential effects on immune system activation, BCG has been reported to neutralize both the herpes virus and the periodontal bacterium [3]. Immunomodulation with BCG has been used in animal models of asthma, irritable bowel disease, multiple sclerosis, and atherosclerosis [4].

Recently, several studies reported that the treatment of NMIBC with BCG instillations reduced the risk of later AD and other dementias by between 27% and 60% [5–7]. However, all these studies about the effect of BCG instillation on the reduction of risks of these diseases have been done by retrospective cohort studies comparing bladder cancer patients with and without this treatment, which implies some inherent bias.

We tested the hypothesis forwarded in the cited papers by investigating the risk of later AD and dementia in a Swedish population-based prospective cohort of patients with NMIBC. We utilized the Bladder Cancer Database Sweden (BladderBaSe) 2.0 with an individually matched comparison cohort and adjusted for comorbidity and socioeconomic factors.

## Materials and methods

### Data source

BladderBaSe is founded on individuals with bladder cancer and cancer of the upper urinary tract consecutively reported to the Swedish National Register for Urinary Bladder Cancer (SNRUBC) [https://statistik.incanet.se/Urinblasecancer/.]. All treatment facilities for urological cancer in Sweden contribute to the register. Entries in SNRUBC are cross-referenced with the Swedish Cancer Register (SCR) for validation and coverage verification. Reporting to the SCR is legally mandated, and active efforts are made to obtain information for patients missing from the SNRUBC. The capture rate for bladder cancer in SNRUBC compared to SCR exceeded 98% from 2017 to 2019 [8]. The SNRUBC contains comprehensive data on tumor characteristics and treatments provided.

BladderBaSe 2.0 is an expanded research database building on BladderBaSe 1.0 [9]. Utilizing the Swedish National Personal Identification Number, this database connects individuals registered in SNRUBC between January 1, 1997, and December 31, 2019, to several national registries.

For this study's purposes, we employed the linkage to the National Patient Register for outpatient and inpatient interventions and diagnoses. In addition to the previous version, BladderBaSe 2.0 also includes a matched comparison cohort sampled from the general population, comprising individuals free from urinary tract cancer at the index case's diagnosis. To construct the comparison cohort, five controls were randomly selected from the general population for each index case, without replacement, on the index case's diagnosis date. Controls were matched based on sex, year of birth, and county of residence. This comparison cohort was linked to the same data sources as the index cases.

### Study population

We applied the following inclusion criteria to identify participants for this study: Patients had to be diagnosed with non-muscle invasive bladder cancer (NMIBC) and have information on BCG as their primary treatment (cases). There were no instances of follow-up loss. The corresponding comparison cohort was included with individual matching preserved. Another control cohort was comprised of NMIBC patients not treated with BCG.

The outcome of interest was the subsequent diagnosis of AD, defined by ICD-10 code F00. We excluded those diagnosed with AD before their NMIBC diagnosis or who received an AD diagnosis within 6 months of their bladder cancer diagnosis. To minimize the risk of missing cases, we also analyzed all subjects with dementia under the following ICD-10 codes: F02, F03, F051, F05, and F06. Subjects with vascular dementia were excluded.

The exposure of interest was treatment with BCG at diagnosis, identified by entries in SNRUBC. Treatment administered during follow-up wasn't registered and thus couldn't be identified.

The study received approval from the Research Ethics Board of Uppsala University, Sweden (Ref no. 2015/277), and the Swedish Ethical Review Authority (Ref no. 2019–03574 and 2020–05123), which waived the requirement for informed consent.

### Statistical methods

Hazard ratios were calculated using univariable Cox proportional hazard regression models with years from bladder cancer diagnosis as the time scale, excluding the first half year from follow-up time to prevent starting follow-up before the event. The Cox model was stratified into strata consisting of each case and its controls. Participants were followed until the event

date (Alzheimer's disease or dementia), or until censoring at the date of death, emigration, or end of follow-up (December 31, 2019), whichever came first. All statistical analyses employed the R statistical package version 4.2.2 [10].

## Results

### Study population

We identified 38,934 patients in the NMIBC cohort (all cases) and a control cohort consisting of 211,045 age-, sex-, and county-matched individuals. Among the former, 17% received BCG after primary diagnosis (cases 1). Table 1 presents the baseline characteristics of these individuals.

The risk analysis encompassed both all forms of dementia and separate consideration of AD diagnoses. During the follow-up period, dementia was diagnosed in 6.1% of cases and 7.4% of controls. For AD specifically, the corresponding rates were 1.5% and 1.9%. The mean follow-up time was shorter for cases compared to controls, likely due to cancer-specific deaths.

### Risk factors

In a comparison between cases treated with BCG (cases 1) and their non-bladder cancer controls, a small but statistically significant reduction in risk was observed for dementia, with a hazard ratio (HR) of 0.88 (95% confidence interval [CI] 0.780–0.991), and for AD with an HR of 0.89 (CI 0.703–1.119). In a subgroup analysis for dementia, an age greater than 75 years yielded an HR of 0.73 (CI 0.616–0.863), and female gender showed an HR of 0.73 (CI 0.552–0.971), both retaining significance. The associations were similar for AD specifically, but statistical significance was not achieved (see Fig 1 and Table 2). To further analyze the impact of

**Table 1. Baseline demographic and clinical characteristics of 6,496 cases (patients with NMIBC who received BCG).**

|  | N = | 6307 |
|---|---|---|
| Age (years) | Mean | 71.3 |
|  | Median | 72.0 |
|  | Q1-Q3 | 66.0–78.0 |
|  | Range | 21–99 |
| Sex | M | 5066 (80.3%) |
|  | F | 1241 (19.7%) |
| Tumor Stage | Ta | 1888 (29.9%) |
|  | Tis | 800 (12.7%) |
|  | T1 | 3619 (57.4%) |
| Grade | G1/LMP | 268 (4.4%) |
|  | G2 | 1913 (31.4%) |
|  | G3-G4/anaplastic | 3916 (64.2%) |
|  | <NA> | 210 |
| CCI | No comorbidity (0) | 3682 (58.4%) |
|  | Mild comorbidity (1) | 1030 (16.3%) |
|  | Intermediate comorbidity (2) | 989 (15.7%) |
|  | Severe comorbidity (>2) | 606 (9.6%) |
| Follow-up time from diagnosis | Mean | 6.0 |
|  | Median | 4.6 |
|  | Q1-Q3 | 2.2–8.8 |
|  | Range | 0–23 |

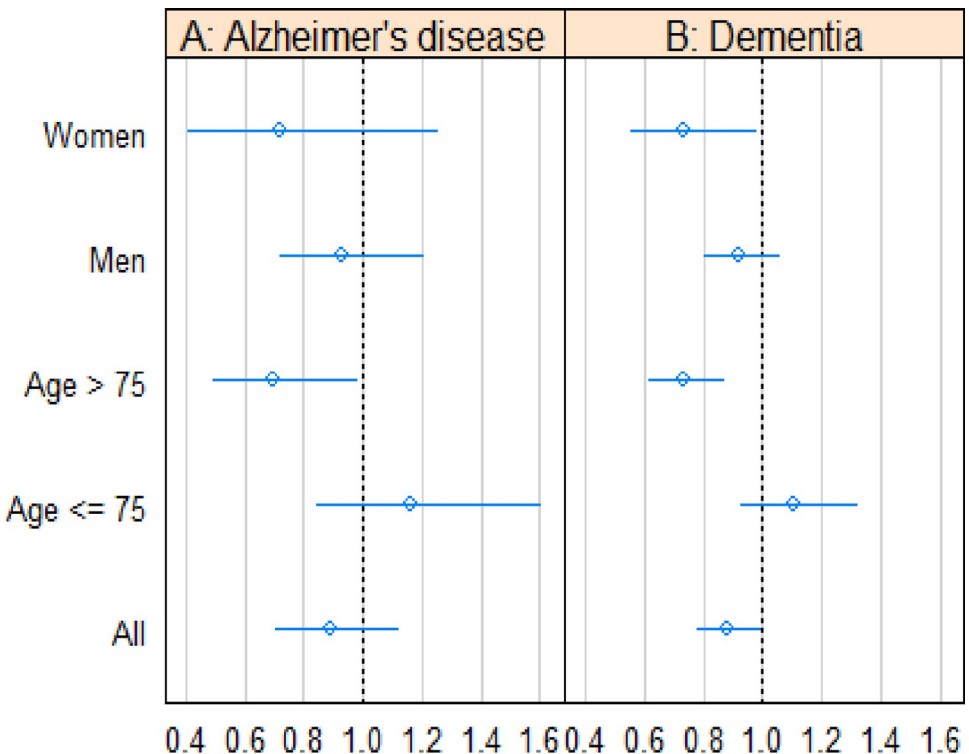

**Fig 1. Associations between BCG-treated individuals and their controls across different subgroups and overall categories.**

extending follow-up to older age, we established a subgroup consisting of patients aged 85 years or older at follow-up. This subgroup included 12,798 patients with a mean age of 79.8. The corresponding results for this subgroup were an HR of 0.87 (CI 0.62–1.23) for AD and an HR of 0.81 (CI 0.69–0.95) for dementia (S1 Table).

To provide a comparative perspective on these risk assessments, we conducted a sensitivity analysis involving cases who did not receive BCG treatment (cases 2) and their respective controls. This analysis unveiled that the risk disparity associated with advanced age persisted

**Table 2. Associations between BCG treated and their controls to be diagnosed with AD or dementia in different subgroups and in all categories.**

| Event | Subset | N (Cases/Controls) | #Events (Cases/Controls) | Mean FU years (Cases/Controls) | HR | 95% CI |
|---|---|---|---|---|---|---|
| Alzheimer's disease | All | 38767 (6060/32707) | 729 (94/635) | 6.17 (5.68/6.26) | 0.89 | 0.703–1.119 |
| | Age < = 75 | 25421 (3925/21496) | 340 (53/287) | 7.09 (6.51/7.19) | 1.16 | 0.842–1.600 |
| | Age > 75 | 13346 (2135/11211) | 389 (41/348) | 4.41 (4.14/4.46) | 0.69 | 0.493–0.976 |
| | Men | 31182 (4862/26320) | 578 (79/499) | 6.10 (5.65/6.18) | 0.93 | 0.722–1.206 |
| | Women | 7585 (1198/6387) | 151 (15/136) | 6.45 (5.80/6.57) | 0.72 | 0.412–1.246 |
| Dementia | All | 38767 (6060/32707) | 2767 (367/2400) | 6.05 (5.59/6.14) | 0.88 | 0.780–0.991 |
| | Age < = 75 | 25421 (3925/21496) | 1220 (179/1041) | 6.99 (6.44/7.09) | 1.11 | 0.930–1.313 |
| | Age > 75 | 13346 (2135/11211) | 1547 (188/1359) | 4.26 (4.02/4.31) | 0.73 | 0.616–0.863 |
| | Men | 31182 (4862/26320) | 2195 (302/1893) | 5.99 (5.55/6.07) | 0.92 | 0.804–1.048 |
| | Women | 7585 (1198/6387) | 572 (65/507) | 6.32 (5.72/6.44) | 0.73 | 0.552–0.971 |

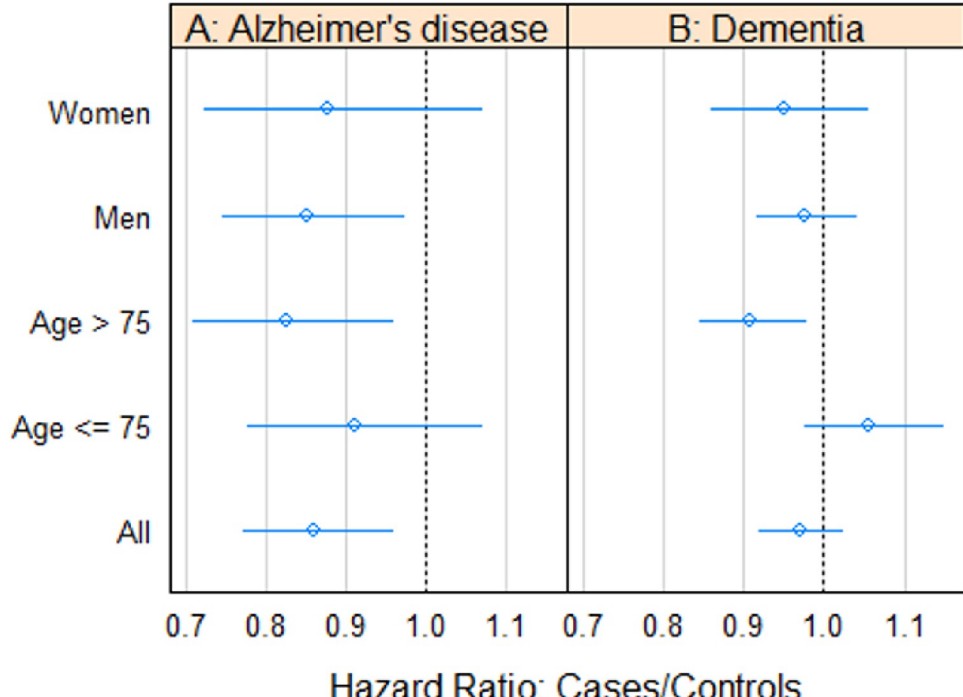

**Fig 2. Associations between non BCG treated and their controls in different subgroups and overall categories.**

within this cohort. Furthermore, it indicated that men exhibited a lower risk of AD compared to women (Fig 2 and Table 3).

Similar to the earlier studies, we also conducted an analysis involving bladder cancer patients who were treated with BCG therapy (cases 1) and those who did not receive BCG therapy (cases 2). The multivariate Cox analysis indicated that BCG-treated patients had a lower risk of dementia compared to those who did not receive BCG, with a hazard ratio (HR) of 0.81 (95% CI 0.71–0.92). The corresponding HR for AD specifically was 0.98 (95% CI 0.75–1.27). Notably, advanced age emerged as a more significant risk modifier for the higher age group, showing an HR of 3.8 (CI 3.44–4.11) for dementia and an HR of 3.1 (CI 2.59–3.73) for AD (S2 Table).

**Table 3. Associations between non BCG treated and their controls to be diagnosed with AD or dementia in different subgroups and overall categories.**

| Event | Subset | N (Cases/Controls) | #Events (Cases/Controls) | Mean FU years (Cases/Controls) | HR | 95% CI |
|---|---|---|---|---|---|---|
| Alzheimer's disease | All | 153340 (23597/129743) | 3495 (446/3049) | 7.27 (6.83/7.35) | 0.86 | 0.772–0.958 |
| | Age < = 75 | 91932 (14103/77829) | 1577 (210/1367) | 8.80 (8.30/8.89) | 0.91 | 0.778–1.067 |
| | Age > 75 | 61408 (9494/51914) | 1918 (236/1682) | 4.99 (4.64/5.05) | 0.83 | 0.711–0.958 |
| | Men | 113931 (17572/96359) | 2466 (313/2153) | 7.15 (6.71/7.23) | 0.85 | 0.748–0.971 |
| | Women | 39409 (6025/33384) | 1029 (133/896) | 7.62 (7.17/7.70) | 0.88 | 0.724–1.067 |
| Dementia | All | 153340 (23597/129743) | 14190 (1947/12243) | 7.11 (6.69/7.19) | 0.97 | 0.919–1.022 |
| | Age < = 75 | 91932 (14103/77829) | 5759 (825/4934) | 8.67 (8.18/8.75) | 1.06 | 0.975–1.146 |
| | Age > 75 | 61408 (9494/51914) | 8431 (1122/7309) | 4.79 (4.48/4.85) | 0.90 | 0.846–0.976 |
| | Men | 113931 (17572/96359) | 10296 (1419/8877) | 7.00 (6.58/7.08) | 0.98 | 0.917–1.040 |
| | Women | 39409 (6025/33384) | 3894 (528/3366) | 7.44 (7.02/7.52) | 0.95 | 0.860–1.053 |

## Discussion

Using data from BladderBaSe 2.0, we compared the effects of BCG instillation on the development of AD and other forms of dementia between patients in a prospective national database with NMIBC who underwent BCG instillation and a matched control group from the general population. Our findings indicated that intravesical BCG treatment was marginally associated with a reduced risk of developing Alzheimer's and dementia. There was suggestive evidence that this modest benefit primarily influenced individuals of advanced age. This latter group exhibited a more than threefold increased overall risk of being diagnosed with dementia and AD among bladder cancer survivors. Additionally, bladder cancer survivors who did not receive BCG demonstrated a decreased risk in comparison to the control group.

Several retrospective analyses have reported a decreased risk in bladder cancer patients treated with BCG compared to those without this therapy. Israeli researchers, in a series of 1371 bladder cancer patients, found that BCG reduced the risk of AD by fourfold relative to controls [7]. Recently, two independent studies in the United States demonstrated similar results. Kim et al. conducted a retrospective study on a cohort of 1,290 racially/ethnically diverse NMIBC patients and found a 60% reduction in the risk of AD and other forms of dementia [5]. In a study based on an administrative dataset comprising 26,584 high-risk NMIBC patients with any exposure to BCG (13,477), a significant 27% lower risk for Alzheimer's disease was observed in comparison to no exposure after adjusting for age, sex, race, T-stage, and Charlson Comorbidity Index [6]. Furthermore, the risk was found to be dose-dependent, with a lower risk associated with higher administered doses. Klinger et al. summarized a retrospective cohort study across three centers, which demonstrated that BCG instillation reduced the risk of developing AD, particularly in men with NMIBC aged older than 75 years [11]. Our results also indicated greater efficacy in older patients, while women demonstrated better outcomes in our context.

There are several differences in our study compared to earlier studies that reported a substantial risk decrease following BCG treatment. Firstly, the study design differs from theirs as their controls were bladder cancer (BC) patients without BCG therapy, while our controls were drawn from the background population. Using BC patients as controls could introduce inherent bias. Their significant risk decrease might be attributed to confounding-by-indication for BCG instillation within bladder cancer patients. Urologists might tend to choose healthier patients for treatment and exclude individuals who are sicker and judged incapable of receiving BCG therapy. Another phenomenon known as 'the healthy individual syndrome' could be at play—healthier, non-smoking individuals who exercise might receive more treatments, while those who neglect health may engage in behaviors harmful to health. As a result, both urologists' and patients' decisions may lead to differences between the BCG and control groups within the NMIBC cohort. An indirect indication of this is that in previous studies, BCG-treated patients were on average 3 years younger and had fewer comorbidities. One might argue that these patients are too young to develop Alzheimer's or dementia during the study period. However, our subgroup analysis focusing on patients aged at least 85 years at follow-up indicates that this wasn't a confounding factor.

The use of BCG in the other reports varied between 25% and 51%, depending on the patient selection in those studies. In Sweden, BCG is recommended for high-risk patients at the time of diagnosis, and our usage rate was 17%. This aligns closely with a prior report that indicated high-risk patients constituted 20% of all NMIBC cases in Sweden [12].

In our dataset, follow-up commenced at the time of primary diagnosis due to the absence of BCG treatment start dates. However, national recommendations suggested initiating

treatment shortly after diagnosis. Our follow-up duration exceeded that of prior reports, with a median of almost five years compared to their three years.

Certain previous studies included details about the dose and duration of treatment, data that we did not have access to. A previous investigation demonstrated a dose-response relationship with BCG, showing a 46% reduction in patients receiving 12 doses or more [13]. The use of various BCG strains could be one of several factors influencing our outcomes.

The subsequent frequency of AD or all other dementia was approximately 8% in other studies, similar to our frequency of 7%. The earlier studies found a similarly decreased risk in other types of dementia, as observed for AD. Consequently, we also included all dementia diagnoses except vascular dementia, which has a distinct pathophysiology.

Diagnoses in Sweden are often made by general practitioners (GPs), and misclassification between dementia subtypes is quite common, as indicated by a validation study of the Swedish registry [13].

Another aging-related disease is cancer, and a recently published meta-analysis of studies on cancer survivors indicated a decreased risk of dementia among them [14]. The pooled relative risk (RR) reached 0.89 for dementia and 0.89 for AD in cancer survivors compared with non-cancer controls. Survivors of colon, leukemia, small intestine, and thyroid cancers experienced the highest decrease, while prostate cancer patients treated with androgen deprivation therapy exhibited a significantly increased risk. Data on larger sets of bladder cancer survivors have not been previously reported to our knowledge. There is a well-known negative impact of both nicotine and tobacco use, but recent nicotine research has pointed to cognitive and neurological benefits linked to smoking [15]. The primary external risk factor for bladder cancer is smoking, and thus, these patients could paradoxically benefit from a lower risk of dementia. Another question is why BCG-treated patients have a lower risk than patients without this therapy. Hypothetically, heavy smokers may have more aggressive disease with larger tumors and higher multiplicity. This could lead urologists to be more eager to initiate BCG therapy despite its potentially serious side effects. Several questions thus remain regarding the associations we found, necessitating further studies in this area.

As in all registry trials, limitations exist in this study. The primary limitation is the use of administrative data to define certain endpoints. The capture rate of quality registers and their representativeness require assessment to ensure the generalizability of register data. Studies of the corresponding prostate cancer register have indicated that information from our quality registers could be generalized to all patients. The absence of randomization could result in some residual confounding. However, our utilization of a background population as controls serves to mitigate this, as demonstrated when we emulated the design of prior studies, revealing a clear benefit from that methodology. We believe our study effectively highlights the challenges arising from the absence of valid controls in comparative studies.

In summary, there could be several explanations for the disparities in results between previous studies and our Swedish study. In our view, the slight reduction in risk compared to a well-matched background population is unlikely to hold significant clinical value. However, the interplay between cancer and the subsequent risk of neuroinflammatory diseases warrants further investigation.

## Conclusion

Our findings suggest that intravesical BCG treatment is not significantly associated with a decreased risk of developing Alzheimer's, but it marginally lowers the risk of all dementia diseases. There is also suggestive evidence that this benefit is primarily pronounced among individuals of advanced age and female gender. As a population-based intervention to mitigate

AD/dementia development, BCG intra-vesical instillation appears to lack substantial efficacy. Notably, bladder cancer survivors who did not receive BCG treatment still exhibited a reduced risk compared to controls.

## Supporting information

**S1 Table. Subgroup analyses of patients follow-up in relevant age.**
(DOCX)

**S2 Table. Multivariate Cox analysis of risk of dementia and AD between BCG treated and non BCG treated patients.**
(DOCX)

## Acknowledgments

We extend our gratitude to the steering committee of Bladderbase, including Staffan Jahnson, Abolfazl Hosseini, Fredrik Liedberg, Truls Gårdmark, Amir Sherif, Viveka Ströck, Karin Söderkvist, Anders Ullen, Christel Häggström, Lars Holmberg, and Firas Aljabery, for their invaluable support.

## Author Contributions

**Conceptualization:** Eugen Wang, Per-Uno Malmström.

**Data curation:** Oskar Hagberg, Per-Uno Malmström.

**Formal analysis:** Oskar Hagberg, Per-Uno Malmström.

**Investigation:** Eugen Wang.

**Methodology:** Oskar Hagberg, Per-Uno Malmström.

**Supervision:** Per-Uno Malmström.

**Writing – original draft:** Eugen Wang.

**Writing – review & editing:** Eugen Wang, Per-Uno Malmström.

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
