## [Decision Letter · Decision Letter 0]

8 May 2023

PONE-D-23-05113The association between BCG treatment in patients with bladder cancer and subsequent risk of developing Alzheimer and other dementia.PLOS ONE

Dear Dr. Wang,

Thank you for submitting your manuscript to PLOS ONE. After careful consideration, we feel that it has merit but does not fully meet PLOS ONE’s publication criteria as it currently stands. Therefore, we invite you to submit a revised version of the manuscript that addresses the points raised during the review process.

We look forward to receiving your revised manuscript.

Kind regards,

Academic Editor

PLOS ONE

Journal Requirements:

"This study was funded by Center for Clinical Research, Sörmland, Uppsala University and Schmekel funder for Urological research."

6. Please amend either the title on the online submission form (via Edit Submission) or the title in the manuscript so that they are identical.

7. Please upload a copy of Supporting Information Table S1 which you refer to in your text on pages 13 and 21. 

**Additional Editor Comments:**

Please revise.

Reviewers' comments:

Reviewer's Responses to Questions

**Comments to the Author**

1. Is the manuscript technically sound, and do the data support the conclusions?

Reviewer #1: Yes

Reviewer #2: Yes

2. Has the statistical analysis been performed appropriately and rigorously? 

Reviewer #1: Yes

Reviewer #2: Yes

3. Have the authors made all data underlying the findings in their manuscript fully available?

Reviewer #1: Yes

Reviewer #2: Yes

4. Is the manuscript presented in an intelligible fashion and written in standard English?

Reviewer #1: Yes

Reviewer #2: Yes

5. Review Comments to the Author

Reviewer #1: Dear Editor

The authors compared Alzheimer and other dementia development among 39,934 bladder patients, 6,496 of them received BCG and control patients. They found that BCG lowers the risk of developing dementia significantly but marginally. The manuscript is well written but there are drawbacks that should be acknowledged.

Comments: 1. Median follow-up was 4.6 years, as compared to 8 years in a previous study (ref#7). Since the median patient’s age was 72 years in this study, there was probably not enough time to develop dementia and to make the difference apparent (median age at diagnosis of Alzheimer’s diseases in ref#7 was 84 years). The authors may study a subgroup of their patients in which follow-up is longer.

2. 6.1% of the study group developed dementia but only 1.5% Alzheimer’s disease. Alzheimer’s disease is responsible for 60-70% of all dementia cases. How can this be explained?

3. Page 15, second paragraph, third line should be ref#7.

Reviewer #2: In their manuscript PONE-D-23-05113, the authors studied the potential effect of BCG instillations and the risk of later developing AD or dementia in a Swedish population-based prospective cohort of patients with non-muscle invasive bladder cancer (NMIBC). The manuscript is organized, well written, and may be suitable for publication. Although this work is very interesting, please see below my comments and questions:

1- Please provide a list of abbreviations, because some abbreviations are mentioned without their full names.

2- It has been shown that adult vaccination against several infections has reduced the risk of developing Alzheimer (AD); how is the difference in BCG vaccine instillations in patients with NMIBC?

3- Accumulating data argue for the beneficial role of BCG to reduce the risk for developing AD. BCG vaccines are manufactured under different conditions across the globe generating divergent formulations, contributing to differences in the ability of these diverse formulations to induce specific and heterologous protection. Could this strains variation explain your results and conclusions that the BCG treatment is not associated with a significant decreased risk of developing Alzheimer but a marginally lower risk for all dementia diseases?

4- In the perspective of this study would it be possible to evaluate if there is any difference in doses exposure to BCG and risk of developing AD or Dementia?

6. PLOS authors have the option to publish the peer review history of their article (what does this mean?). If published, this will include your full peer review and any attached files.

Reviewer #1: No

Reviewer #2: No

---

## [Author Response · Author response to Decision Letter 0]

28 Jun 2023

Here is a point-by-point response to the reviewers’ comments and concerns.

Reviewers comments and concerns our responses and changes

Editor 

General comments 

1. Please ensure that your manuscript meets PLOS ONE's style requirements OK

2. We note that the grant information you provided in the ‘Funding Information’ and ‘Financial Disclosure’ sections do not match. OK, we will mention our financial disclosures as following: 

"This study was funded by Center for Clinical Research, Sörmland, Uppsala University and Schmekel funder for Urological research."

"This study was funded by Center for Clinical Research, Sörmland, Uppsala University and Schmekel funder for Urological research."

Please include this amended Role of Funder statement in your cover letter; The funders had no role in study design, data collection and analysis, decision to publish, or preparation of the manuscript

4. In your Data Availability statement, you have not specified where the minimal data set underlying the results described in your manuscript can be found. PLOS defines a study's minimal data set as the underlying data used to reach the conclusions drawn in the manuscript and any additional data required to replicate the reported study findings in their entirety. All PLOS journals require that the minimal data set be made fully available. For more information about our data policy, please see

 Data used in the present study was extracted from the research database BladderBaSe, which is based on the Swedish National Registry of Urinary Bladder Cancer (SNRUBC) and linkage to several national health-data registers. The data cannot be shared publicly because the individual-level data contain potentially identifying and sensitive patient information and cannot be published due to legislation and ethical review restrictions (https://etikprovningsmyndigheten.se). Use of the data from national health-data registers is further restricted by the Swedish Board of Health and Welfare (https://www.socialstyrelsen.se/en/) and Statistics Sweden (https://www.scb.se/en/) which are Government Agencies providing access to the linked healthcare registers.

The data in in BladderBaSe is partly available in annual reports from the Swedish National Registry of Urinary Bladder Cancer (SNRUBC) and are also accessible online at https://statistik.incanet.se/urinblasecancer/. Researchers can propose and apply for data and studies in BladderBaSe or SNRUBC using a standardized form. After approved application, the project data administrators can upload study-specific files with selected variables to a server for statistical analysis through remote access.

De-identified data can be available to researchers after application to the BladderBaSe Steering Committee.

6. Please amend either the title on the online submission form (via Edit Submission) or the title in the manuscript so that they are identical.

 OK

7. Please upload a copy of Supporting Information Table S1 which you refer to in your text on pages 13 and 21. 

 OK

Additional Editor Comments:

Reviewer #1: 

1. Median follow-up was 4.6 years, as compared to 8 years in a previous study (ref#7). Since the median patient’s age was 72 years in this study, there was probably not enough time to develop dementia and to make the difference apparent (median age at diagnosis of Alzheimer’s diseases in ref#7 was 84 years). The authors may study a subgroup of their patients in which follow-up is longer.

 It is correct that we had a shorter follow-up but the mean age was three years more in our study group.

To analyse if the shorter follow-up had an impact we established one subgroup who were 85 years or older at follow-up. As far as we can see, our conclusions will not be changed even in this subgroup. 

This information is now added in the results.

2. 6.1% of the study group developed dementia but only 1.5% Alzheimer’s disease. Alzheimer’s disease is responsible for 60-70% of all dementia cases. How can this be explained?

 Our data from register was like as we presented, which may be because doctors (especially general practitioners from Health center) did not give detail diagnosis with Alzheimers, but only generally diagnosis with dementia. In discussion, we mentioned: The diagnosis in Sweden is often made by the GP ( general practitioners) and misclassification between dementia subtypes is quite common according to a validation study of the Swedish registry.

3. Page 15, second paragraph, third line should be ref#7.

 Ok

Reviewer #2: 

1- Please provide a list of abbreviations, because some abbreviations are mentioned without their full names.

 List of abbreviations: 

AD – Alzheimer´s disease

BCG - Bacillus Calmette-Guerin

BladderBaSe - Bladder Cancer Data Base Sweden

CI – confidence interval

G1/LMP: Bladder cancer Grade 1 or Low Malignant Potential

GP - general practitioner 

HR – hazard ratio

ICD – International classification of diseases 

NMIBC - non-muscle invasive bladder cancer

Q1-Q3; Inter quartile range, the range between the first quartile (Q1) and the third (Q3)

SNRUBC – Swedish national register of urinary bladder cancer

2- It has been shown that adult vaccination against several infections has reduced the risk of developing Alzheimer (AD); how is the difference in BCG vaccine instillations in patients with NMIBC? We do not fully understand this question but it is clear that BCG bladder instillations leads to a systemic immune response with for example fever in some patients and getting a positive PPD skin test. 

3- Accumulating data argue for the beneficial role of BCG to reduce the risk for developing AD. BCG vaccines are manufactured under different conditions across the globe generating divergent formulations, contributing to differences in the ability of these diverse formulations to induce specific and heterologous protection. Could this strains variation explain your results and conclusions that the BCG treatment is not associated with a significant decreased risk of developing Alzheimer but a marginally lower risk for all dementia diseases? Yes, using different BCG strains could be one of many factors to influence our results, which we will mention in our discussion. 

4- In the perspective of this study would it be possible to evaluate if there is any difference in doses exposure to BCG and risk of developing AD or Dementia?

 In discussion we wrote that one limitation was that dose and duration of the treatment, was data we did not have available. 

6. PLOS authors have the option to publish the peer review history of their article (what does this mean?). If published, this will include your full peer review and any attached files. OK.

---

## [Decision Letter · Decision Letter 1]

24 Jul 2023

PONE-D-23-05113R1The association between BCG treatment in patients with bladder cancer and subsequent risk of developing Alzheimer and other dementia.

- A Swedish nationwide cohort study from 1997 to 2019.PLOS ONE

Dear Dr. Wang,

Thank you for submitting your manuscript to PLOS ONE. After careful consideration, we feel that it has merit but does not fully meet PLOS ONE’s publication criteria as it currently stands. Therefore, we invite you to submit a revised version of the manuscript that addresses the points raised during the review process.

Please revise.

We look forward to receiving your revised manuscript.

Kind regards,

Academic Editor

PLOS ONE

Journal Requirements:

Reviewers' comments:

Reviewer's Responses to Questions

**Comments to the Author**

1. If the authors have adequately addressed your comments raised in a previous round of review and you feel that this manuscript is now acceptable for publication, you may indicate that here to bypass the “Comments to the Author” section, enter your conflict of interest statement in the “Confidential to Editor” section, and submit your "Accept" recommendation.

Reviewer #1: All comments have been addressed

Reviewer #2: All comments have been addressed

2. Is the manuscript technically sound, and do the data support the conclusions?

Reviewer #1: No

Reviewer #2: Yes

3. Has the statistical analysis been performed appropriately and rigorously? 

Reviewer #1: Yes

Reviewer #2: Yes

4. Have the authors made all data underlying the findings in their manuscript fully available?

Reviewer #1: Yes

Reviewer #2: Yes

5. Is the manuscript presented in an intelligible fashion and written in standard English?

Reviewer #1: Yes

Reviewer #2: Yes

6. Review Comments to the Author

Reviewer #1: Dear Editor

Thank you for your letter and for the response of the authors.

Previous studies with long follow-up showed significant decrease in the risk of Alzheimer's disease in BCG treated patients. The current study with a short follow-up found only a marginal decrease. One may figure that with longer follow-up the marginal benefit will be more pronounced.

The explanation why only 1.5% patient were diagnosed with Alzheimer's disease (out of 6.1% patients with dementia), due to the poor capacity of the doctors, cast shadow on all their diagnoses.

Reviewer #2: In the present revised manuscript PONE-D-23-05113R1, the authors have addressed answers to questions and they have incorporate changes to reflect most of the suggestions. Their article may be suitable for publication in the PLOS journal.

7. PLOS authors have the option to publish the peer review history of their article (what does this mean?). If published, this will include your full peer review and any attached files.

Reviewer #1: No

Reviewer #2: No

---

## [Author Response · Author response to Decision Letter 1]

1 Aug 2023

Following is our answer for comment from reviewer 1: 

Reviewers comments and concerns: 

Previous studies with long follow-up showed significant decrease in the risk of Alzheimer's disease in BCG treated patients. The current study with a short follow-up found only a marginal decrease. One may figure that with longer follow-up the marginal benefit will be more pronounced.

Our answer:

The median follow-up was 8 years in one study (ref 7) and about 3 years in the other previous studies (ref 5 and 6) compared to 4.6 years in our study. Both these later studies showed a substantial decrease in risk as opposed to our marginal decrease despite similar follow-up length.

To further address the reviewers concern regarding the follow-up time we conducted a subgroup analysis with subjects followed until using the number of years from year of diagnosis to 2019. If this number plus the age of the patient is at least 85, the patients are kept in the analysis. I.e., only patients who could be under risk at age 85 or older are kept in the analysis. Then 12798 patients with a mean age of 79.8 at diagnosis remain with a median follow-up time of about 5 years. The HR was then 0.87 and 0.81 for AD and dementia respectively. This was in the same range as for the whole study group (0.89 and 0.88) with shorter follow up. As more details to explain this point, we will add one more table (Table S2) in support information. 

Reviewers comments and concerns: 

The explanation why only 1.5% patient were diagnosed with Alzheimer's disease (out of 6.1% patients with dementia), due to the poor capacity of the doctors, cast shadow on all their diagnoses.

Our Answer: 

The first referred study (ref 5) presented results separately for AD and other dementia. The AD represented 24% of all dementias as opposed to 25 % in our study. The second referred study (ref 6) reported 6 % with AD as sole diagnosis and 36 % with AD coupled to other types of dementia. The third study (ref 7) did not analyse other dementia than AD. 

Reviewer 2 had no comments.

---

## [Decision Letter · Decision Letter 2]

30 Aug 2023

PONE-D-23-05113R2The association between BCG treatment in patients with bladder cancer and subsequent risk of developing Alzheimer and other dementia.- A Swedish nationwide cohort study from 1997 to 2019.PLOS ONE

Dear Dr. Wang,

Thank you for submitting your manuscript to PLOS ONE. After careful consideration, we feel that it has merit but does not fully meet PLOS ONE’s publication criteria as it currently stands. Therefore, we invite you to submit a revised version of the manuscript that addresses the points raised during the review process.

Please revise.

We look forward to receiving your revised manuscript.

Kind regards,

Academic Editor

PLOS ONE

Journal Requirements:

Reviewers' comments:

Reviewer's Responses to Questions

**Comments to the Author**

1. If the authors have adequately addressed your comments raised in a previous round of review and you feel that this manuscript is now acceptable for publication, you may indicate that here to bypass the “Comments to the Author” section, enter your conflict of interest statement in the “Confidential to Editor” section, and submit your "Accept" recommendation.

Reviewer #1: All comments have been addressed

Reviewer #2: All comments have been addressed

2. Is the manuscript technically sound, and do the data support the conclusions?

Reviewer #1: Yes

Reviewer #2: Yes

3. Has the statistical analysis been performed appropriately and rigorously? 

Reviewer #1: Yes

Reviewer #2: Yes

4. Have the authors made all data underlying the findings in their manuscript fully available?

Reviewer #1: Yes

Reviewer #2: Yes

5. Is the manuscript presented in an intelligible fashion and written in standard English?

Reviewer #1: Yes

Reviewer #2: Yes

6. Review Comments to the Author

Reviewer #1: None. Congratulations to the authors. They answered all the questions. Hope the production will be adequate to

Reviewer #2: The reviewed manuscript entitled " The association between BCG treatment in patients with bladder cancer and subsequent risk of developing Alzheimer and other dementia.- A Swedish nationwide cohort study from 1997 to 2019",referenced under the number PONE-D-23-05113-R2, has been improved after implementing the reviewer’s suggestions, and may be acceptable for publication. Thank you for accurate response to suggestions.

Minor remarks:

The paper should be reviewed by an English native speaker. For example, the following sentences may need to be rewritten to avoid any ambiguity:

-Results, Risk factors: “To further analyse age at follow-up had an impact we established…”

- Discussion: the sentence starting by « One may argue that patients are to young to develop…”

7. PLOS authors have the option to publish the peer review history of their article (what does this mean?). If published, this will include your full peer review and any attached files.

Reviewer #1: No

Reviewer #2: No

---

## [Author Response · Author response to Decision Letter 2]

2 Sep 2023

The paper has been now reviewed by an English native speaker, who made changes to a text without altering its meaning, according to the recommendation from reviewer #2.

---

## [Decision Letter · Decision Letter 3]

14 Sep 2023

The association between BCG treatment in patients with bladder cancer and subsequent risk of developing Alzheimer and other dementia.- A Swedish nationwide cohort study from 1997 to 2019.

PONE-D-23-05113R3

Dear Dr. Wang,

We’re pleased to inform you that your manuscript has been judged scientifically suitable for publication and will be formally accepted for publication once it meets all outstanding technical requirements.

Kind regards,

Academic Editor

PLOS ONE

Additional Editor Comments (optional):

If possible, please revise the Title to make it more concise whereas informative. 

Reviewers' comments:

Reviewer's Responses to Questions

**Comments to the Author**

1. If the authors have adequately addressed your comments raised in a previous round of review and you feel that this manuscript is now acceptable for publication, you may indicate that here to bypass the “Comments to the Author” section, enter your conflict of interest statement in the “Confidential to Editor” section, and submit your "Accept" recommendation.

Reviewer #1: All comments have been addressed

Reviewer #2: All comments have been addressed

2. Is the manuscript technically sound, and do the data support the conclusions?

Reviewer #1: Yes

Reviewer #2: Yes

3. Has the statistical analysis been performed appropriately and rigorously? 

Reviewer #1: Yes

Reviewer #2: Yes

4. Have the authors made all data underlying the findings in their manuscript fully available?

Reviewer #1: Yes

Reviewer #2: Yes

5. Is the manuscript presented in an intelligible fashion and written in standard English?

Reviewer #1: Yes

Reviewer #2: Yes

6. Review Comments to the Author

Reviewer #1: The only issue left was the linguistic review which has been addressed.

I hope that the manuscript will provide an important perspective to the growing interest in the association between the immune system and Alzheimer's disease.

Reviewer #2: The reviewed manuscript referenced under the number PONE-D-23-05113-R3, has been improved after implementing the reviewer’s suggestions, and may be acceptable for publication. Thank you for accurate response to suggestions.

7. PLOS authors have the option to publish the peer review history of their article (what does this mean?). If published, this will include your full peer review and any attached files.

Reviewer #1: No

Reviewer #2: No

---

## [Editor Report · Acceptance letter]

6 Dec 2023

PONE-D-23-05113R3 

The association between BCG treatment in patients with bladder cancer and subsequent risk of developing Alzheimer and other dementia. 
- A Swedish nationwide cohort study from 1997 to 2019. 

Dear Dr. Wang:

I'm pleased to inform you that your manuscript has been deemed suitable for publication in PLOS ONE. Congratulations! Your manuscript is now with our production department. 

Kind regards, 

on behalf of

Dr. Robert Jeenchen Chen 

Academic Editor

PLOS ONE